

# Synergy between roads and disturbance favour *Bromus tectorum* L. invasion

Karina L. Speziale[1], Agustina di Virgilio[1,2], Maria N. Lescano[3], Gabriela Pirk[3] and Jorgelina Franzese[3]

[1] Grupo de Investigaciones en Biología de la Conservación, Departamento de Ecología, Laboratorio Ecotono, INIBIOMA(CONICET-UNCOMA), Bariloche, Argentina
[2] Grupo de Ecología Cuantitativa, INIBIOMA, CONICET-UNCOMA, Bariloche, Río Negro, Argentina
[3] Departamento de Ecología, Laboratorio Ecotono, INIBIOMA (CONICET-UNCOMA), Bariloche, Argentina

## ABSTRACT

**Background**. Global change produces pervasive negative impacts on biodiversity worldwide. Land use change and biological invasions are two of the major drivers of global change that often coexist; however, the effects of their interaction on natural habitats have been little investigated. In particular, we aimed to analyse whether the invasion of an introduced grass (*Bromus tectorum*; cheatgrass) along roads verges and the disturbance level in the natural surrounding habitat interact to influence the degree of *B. tectorum* invasion in the latter habitats in north-western Patagonia.

**Methods**. Along six different roads, totalling approximately 370 km, we set two 50 m × 2 m sampling plots every 5 km (73 plots in total). One plot was placed parallel to the road (on the roadside) and the other one perpendicular to it, towards the interior of the natural surrounding habitat. In each plot, we estimated the *B. tectorum* plant density in 1 m$^2$ subplots placed every 5 m. In the natural habitat, we registered the vegetation type (grassy steppe, shrub-steppe, shrubland, and wet-meadow) and the disturbance level (low, intermediate, and high). Disturbance level was visually categorized according to different signs of habitat degradation by anthropogenic use.

**Results**. *B. tectorum* density showed an exponential decay from roadsides towards the interior of natural habitats. The degree of *B. tectorum* invasion inside natural habitats was positively related to *B. tectorum* density on roadsides only when the disturbance level was low. Shrub-steppes, grassy steppes and shrublands showed similar mean density of *B. tectorum*. Wet-meadows had the lowest densities of *B. tectorum*. Intermediate and highly disturbed environments presented higher *B. tectorum* density than those areas with low disturbance.

**Discussion**. Our study highlights the importance of the interaction between road verges and disturbance levels on *B. tectorum* invasion in natural habitats surrounding roads of north-western Patagonia, particularly evidencing its significance in the invasion onset. The importance of invasion in road verges depends on disturbance level, with better conserved environments being more resistant to invasion at low levels of *B. tectorum* density along road verges, but more susceptible to road verges invasion at higher levels of disturbance. All the habitats except wet-meadows were invaded at a similar degree by *B. tectorum*, which reflects its adaptability to multiple habitat conditions. Overall, our work showed that synergies among global change drivers impact native environments favouring the invasion of *B. tectorum*.

Corresponding author
Karina L. Speziale,
karina.speziale@comahue-conicet.gob.ar

# INTRODUCTION

In the Anthropocene era in which we are now living, human activities impacting on biodiversity are so widespread that they are collectively known as anthropogenic global change (*Vitousek et al., 1996*; *Sala et al., 2000*). Global change produces pervasive negative impacts on biodiversity from all Earth's ecosystems as consequences of land use change, biological invasions, climate change, overexploitation and alteration of bio-geo-chemical cycles (*Sala et al., 2000*). The loss of ecosystem functions and services due to biodiversity changes is producing economic impacts via the reduction of food sources, fuel, structural materials, or by changing community composition and vulnerability to invasion (*Chapin III et al., 2000*). According to Sala and colleagues (*2000*) and Vitousek and colleagues (*1996*) two of the major global change drivers threatening biodiversity are currently land use change (e.g., habitat transformation into roads or grazing by domestic animals) and biological invasions (*Vitousek et al., 1996*; *Dukes & Mooney, 1999*; *Sala et al., 2000*). Of particular concern is the simultaneous action of separate processes that have a greater total effect than the sum of individual effects alone i.e., synergy among global change drivers (*Dukes & Mooney, 1999*; *Brook, Sodhi & Bradshaw, 2008*) which is rarely studied (*Didham et al., 2005*).

Among land use change drivers, roads are still dominating human movements, carrying along them unwanted biological organisms and favouring their long distance dispersal (*Von der Lippe & Kowarik, 2007*; *Strano et al., 2018*). Roads are a major contributor for the spread of introduced plant species (*Forman & Alexander, 1998*; *Gelbard & Belnap, 2003*; *Ibisch et al., 2016*). Habitats adjacent to roads are often more homogeneous than natural or semi-natural habitats, as they are subject to increased disturbance and decreased competition compared to more distant natural habitats (*Forman & Alexander, 1998*; *Spellerberg, 1998*). As a consequence, road verges may sometimes harbor short lived, fast growing species which allocate a large proportion of their photosynthesis products to seed output (*Frenkel, 1977*), many of which are introduced species (*Gelbard & Belnap, 2003*; *Trombulak & Frissell, 2000*). The linear arrangement of roads, their maintenance works and ordinary traffic, increase plant species dispersal along roadsides, particularly of introduced species (*Forman, 2003*; *Lembrechts et al., 2016*; *Dainese et al., 2017*; *Rew et al., 2018*).

Once established in roadside habitats, non-native plant species may spread into surrounding environments (*Seipel et al., 2012*). For instance, introduced species richness is lower in interior habitats respect to roads in Glacier National Park (*Tyser & Worley, 1992*), in California (*Frenkel, 1977*), and in south-eastern Ohio (*Christen & Matlack, 2009*) in USA. Both global change drivers, roads and biological invasions, may in addition combine with other land use change drivers within interior habitats pointing at complex interactions as the mechanism producing ecosystem alterations. Among the ones favouring the spread of introduced species within native communities, grazing by domestic animals,

and other disturbances are included, as well as extreme weather conditions due to climate change (*Brandt & Rickard, 1994*; *Davis, Grime & Thompson, 2000*; *Bradley, 2009*). Of them, overgrazing often favours the invasion of Eurasian grasses possibly due to their longer co-evolution with ungulate grazers turning these grasses more resistant to trampling and grazing (*Mack, 1986*; *Tyser & Worley, 1992*). These potential interactions and synergies among drivers may be highly important in determining the actual impact on biodiversity but yet, they have been little studied (*Didham et al., 2007*).

*Bromus tectorum* L. (cheatgrass-downy brome) is a winter annual grass species of Eurasian origin. It is considered invasive in 11 countries, being USA the one with most records (ISSG 2017). In this country it is commonly found along roads and disturbed areas (*Hulbert, 1955*; *Gelbard & Belnap, 2003*). Roads act as corridors, particularly favouring *B. tectorum* cover along paved roads (*Gelbard & Belnap, 2003*). Seeds are released within the first weeks after ripening at the end of spring. Long distance dispersal is driven by positive interactions with grazers, which, together with its higher competition efficiency with native perennial grasses favour the invasion process (*Hulbert, 1955*). Unusual phenotypical plasticity and greater efficiency in water and/or nitrogen use probably enable *B. tectorum* to be more suited to frequent disturbance than native species (*Mack & Pyke, 1983*; *Rice et al., 1992*; *Lowe, Lauenroth & Burke, 2003*). However, undisturbed sagebrush habitats for example, are resistant to *B. tectorum* invasion (*Lavin et al., 2013*). Thus, the conservation status of each environment may then determine the magnitude of the invasion (*Rickard & Vaughan, 1988*; *Bradford & Lauenroth, 2006*).

*B. tectorum* has been recently described as an invasive species in Patagonia (*Speziale, Lambertucci & Ezcurra, 2014*) and its distribution is increasing since the first record in 1937 (*Biganzoli, Larsen & Rolhauser, 2013*). Particularly, in north-western Patagonia it is widespread along roads (KS personal observation). Like the USA, north-western Patagonia presents climatic conditions that favours its persistence, with cattle raising areas the most invaded ones (*Veblen et al., 1992*; *Bradford & Lauenroth, 2006*; *Speziale, Lambertucci & Ezcurra, 2014*).

In this work we sampled *B. tectorum* along road verges and in their close surrounding environments in north-western Patagonia (Argentina), part of the area where the species is described as invader. The comparison of roadsides and natural habitats enabled us to test the invasion degree in relation to the density of *B. tectorum* at road verges. Additionally, it also allowed us to understand whether the impact of roads interacts with other land-use changes to determine the invasion degree of surrounding environments. We aimed to analyse whether the density of *B. tectorum* along road verges and the disturbance level in the surrounding natural habitats favour the invasion. We hypothesized that the density of *B. tectorum* on the roadsides together with the level of disturbance within the surrounding landscape influence the density of *B. tectorum* in this latter habitat. We predict that areas with higher density of *B. tectorum* on the roadsides will record the highest densities of *B. tectorum* within surrounding environments when disturbance is high.

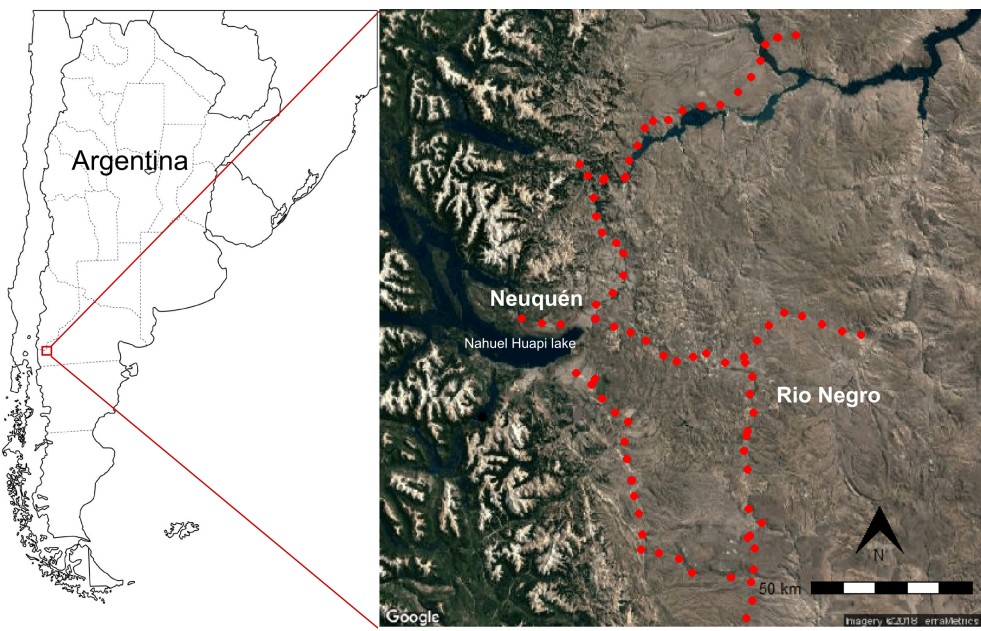

**Figure 1** **Study area in northwestern Patagonia.** Study was conducted along roads of Neuquén and Rio Negro provinces in Argentina. Red dots show sampling plot location. Map data Google, SIO, NOAA, US Navy, GEBCO, US Dept of State Geographer, Image Landsat/Copernicus.

## MATERIALS & METHODS

### Study area

We conducted our fieldwork on the extra-Andean zones in Argentina within the Patagonian sub-region (Fig. 1). The area is subject to a pronounced environmental gradient (*Barros et al., 1983*) with four distinct dominant physiognomic units: steppes, shrublands, shrub-steppes and wet-meadows (Fig. 1; *Paruelo et al., 1998a*). Steppes are dominated by perennial grasses as *Poa ligularis*, *Festuca pallescens,* and *Pappostipa speciosa*. Shrublands are dominated by shrubs as *Mulinum spinosum*, *Schinus patagonica*, and *Anarthrophyllum rigidum,* among others. Shrub-steppes are composed of perennial grasses and include low to medium height shrubs as *M. spinosum*, *A. rigidum*, *Colliguaja intergerrima,* and *Adesmia* spp. Wet-meadows (mallines) are diverse edaphic communities in humid depressions dominated by *Juncus* spp, *Distichlis* spp. and other humid-adapted species. In this region, the southern Andes act as a barrier to the humid westerlies causing a greater amount of precipitation in the Andean Cordillera compared with a few 150 km to the east (*Barros et al., 1983*; *Paruelo et al., 1998b*). Precipitation mainly occurs in autumn, and winter ranging from about 800 mm/year in the west to 300 mm/year to the east of our study area. To the west, the area includes shrublands within Nahuel Huapi National Parks reserve, where lands are privately owned. Within the region tourism, forestry and cattle and sheep rearing are the main economic activities. Among the important disturbances in the area are anthropogenic fires and grazing by introduced animals (*Veblen et al., 1992*).

## Sampling design

We travelled the area along six different roads around 130 km on paved roads and 240 km on unpaved roads. In each road, we set sampling plots every 5 km, with a total of 73 plots (Fig. 1). Each plot consisted of two sub-plots of 50 m long and 2 m wide. One sub-plot was placed parallel to the road and the other perpendicular to the centre of the first one, towards the interior of the field; starting by the fencing when present. If not, the sub-plot perpendicular to the road started when a vegetation change was observed from vegetation disturbed by road or road maintenance to more pristine vegetation. In each plot, we located a 1 m$^2$ frame subdivided into a 100-celled -grid every 5 m and we estimated density in categories by counting the number of *B. tectorum* plants: 0, no individuals; 1, from one to 15 individuals; 2, from 16 to 25 individuals; 3, from 26 to 50 individuals; 4, from 51 to 75 individuals; 5, more than 75 individuals. For statistical analysis (see below), this density level was expressed as the approximate density using the maximum number of individuals plants listed above per m$^2$ (hereafter *B. tectorum* density). When the species was not present up to the first 50 m to the interior of the habitat we considered it absent without sampling any further. But when *B. tectorum* was still present, we kept walking 50 m more to register the maximum distance where *B. tectorum* could be found and assigned a category of density. When we still found *B. tectorum* beyond those 100 m we recorded its presence as "more than 100 m". This resulted in distances sampled that ranged from 0 to 100 m away from roads. Additionally, we registered the vegetation type, and disturbance level. For vegetation type we used four categories: grassy steppe, shrub-steppe, shrubland, and wet-meadow. Disturbance level was visually categorized in low (no clear sign of disturbance), intermediate (low percentage of bare soil, few signs of herbivore damage to soil and vegetation and few herbivore feces), and high (high percentage of bare soil, signs of soil disturbance due to grazing or human activities, presence of species common in degraded areas, top soil removed, and herbivore feces). All sampling sites were areas used for extensive livestock production with a number of samples for each habitat type of: $n = 26$ for grassy steppes; $n = 38$ for shrub-steppes; $n = 8$ for shrublands; and $n = 2$ for wet-meadows. Elevation ranged from 705 to 1,240 m asl (Sup Mat). Field work was approved by the National Park Administration (project number: 1526).

## Analyses

To assess the effects of road verges on the density of *B. tectorum* in surrounding environments, we fitted a generalized mixed effects model with Poisson distribution (Supplemental Information 1). Our response variable was the density of *B. tectorum* per 1 m$^2$ plot in the surrounding environments and the explanatory variable the density of *B. tectorum* in road verges. We included co-variables to assess the influence of other potential variables affecting the results. For this we included the distance (in meters) from the road verge, the level of disturbance and vegetation type of the environment. The quantitative co-variables were centred and standardized to make their coefficients comparable and to ease computational estimations. Following our hierarchical sampling design, we used Road and Sampling Transect as nested random effects. We considered intermediate disturbance levels and shrubland vegetation as reference group and included an interaction term

**Table 1  Model's fixed effects coefficients.** Expected values in log scale, standard errors (SE), degrees of freedom (DF), $t$-values, 95% confidence intervals (95% CI), and $p$-values for the model's fixed effects. $\beta_0$ is the model's intercept, which includes intermediate disturbance level and shrubland vegetation as reference group. $\beta_1$ and $\beta_2$ are the regression coefficients for high and low levels of field disturbance respectively. $\beta_3$ represents the effect of *B. tectorum* density (plants*m$^2$) at road verges on field density. $\beta_4$, $\beta_5$ and $\beta_6$ show the effects of grassland, wetland and shrubland vegetation on *B. tectorum* density in the field, respectively. $\beta_7$ represents the effect of distance from verges (in meters) on the density of *B. tectorum* in the field. $\beta_8$ and $\beta_9$ are the interaction terms between *B. tectorum* density on road verges and the level of disturbance of the field.

| Parameters (log scale) | Estimated | SE | DF | $t$-value | 95% CI Lower | 95% CI Upper | $p$-value |
|---|---|---|---|---|---|---|---|
| $\beta_0$: Intercept | 2.13 | 0.19 | 859 | 11.39 | 1.76 | 2.49 | 0.001* |
| $\beta_1$: High disturbance level | 0.31 | 0.29 | 859 | 1.07 | −0.27 | 0.90 | 0.288 |
| $\beta_2$: Low disturbance level | −1.15 | 0.43 | 859 | −2.66 | −2.01 | −0.29 | 0.010* |
| $\beta_3$: *Bt* density in verge | 1.01 | 0.14 | 859 | 7.46 | 0.74 | 1.28 | 0.001* |
| $\beta_4$: Grassland | −0.21 | 0.28 | 859 | −0.75 | −0.76 | 0.34 | 0.454 |
| $\beta_5$: Wetland | −0.95 | 0.10 | 859 | −2.28 | −2.07 | −1.57 | 0.002* |
| $\beta_6$: Shrubland | −0.25 | 0.46 | 859 | −0.54 | −1.16 | 0.66 | 0.591 |
| $\beta_7$: Distance from verge | −0.10 | 0.03 | 859 | −3.31 | −0.17 | −0.04 | 0.001* |
| $\beta_8$: *Bt* verges * High disturb | 0.04 | 0.03 | 859 | 1.29 | −0.02 | 0.10 | 0.201 |
| $\beta_9$: *Bt* verges * Low disturb | 1.59 | 0.11 | 859 | 14.02 | 1.37 | 1.81 | 0.001* |

**Notes.**

Statistically significant $p$-values are marked with *, and Bt is an abbreviation for *B. tectorum*.

between level of disturbance and the average density of *B. tectorum* in the road verges. All analyses were performed in R (*R Core Team, 2016*). Due to the spatial nature of our data, we checked spatial autocorrelation by constructing a correlogram, using the function spline.correlog( ) from ncf package (*Bjornstad, 2009*). After fitting the model that included the spatial correlation term, we checked for spatial autocorrelation in the normalized model's residuals using the acf( ) function. The final model, selected by the lowest AIC, was a linear model with an interaction term between the density of *B. tectorum* in road verges and the level of disturbance on the surrounding environment. Models fitting was performed using the gamm( ) function from mgcv package (*Wood, 2011*).

## RESULTS

We found that the density of *B. tectorum* in road verges interacted with the disturbance in the surrounding environment to determine its density in these latter habitats (Table 1). When the contiguous environment was more conserved (i.e., lower levels of disturbance), the influence of road verges on the density of *B. tectorum* inside the surrounding environment was high. Instead, when this environment was moderate or highly disturbed the density of *B. tectorum* in these areas did not depend on road verges densities. This result shows that in environments with low disturbance levels, the density of *B. tectorum* increases as their density in road verges increases.

We also found that *B. tectorum* density decreases when we moved away from road verges towards the interior of surrounding environment (Table 1). This negative effect of distance
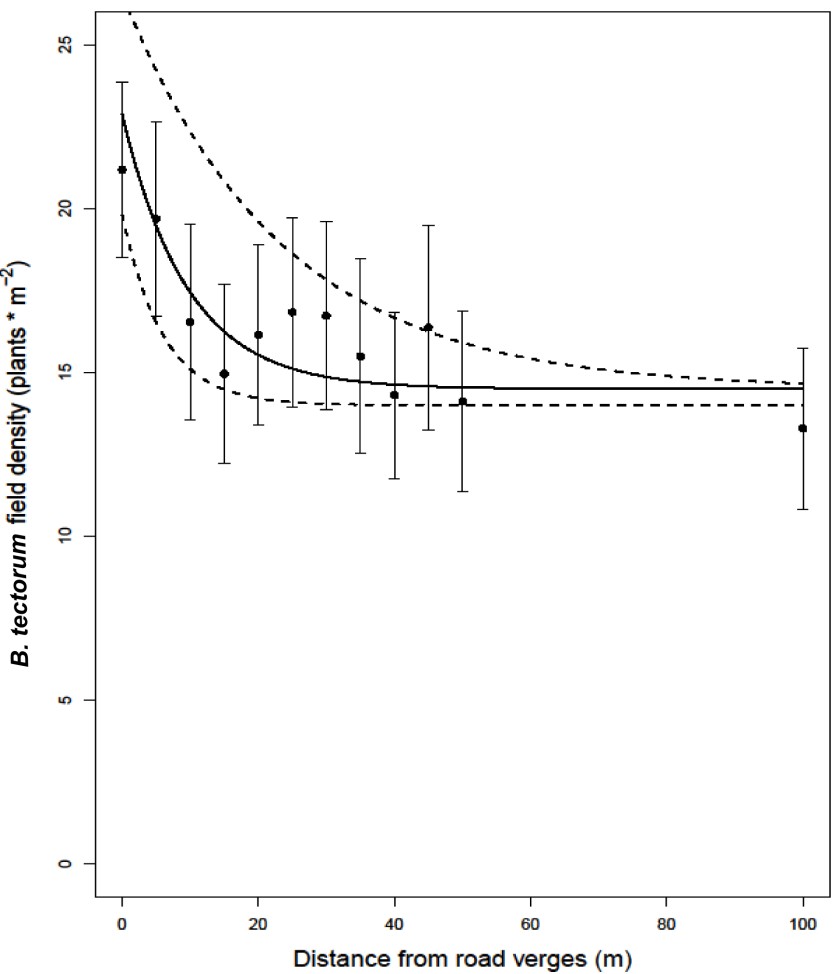

**Figure 2 Exponential decay of *B. tectorum* density with distance from road verges.** The black dots represent the mean observed values of density at each measured distance, and the bars are the standard errors. The solid line is the expected decay obtained from model fit, and dashed lines are the 95% confident intervals.

from road verges on *B. tectorum* density showed an exponential decay of density when distance increases, at a rate of 0.9 plants per meter (Fig. 2).

Results from the model fit confirmed the observations and raw data that shrub-steppes, grassy steppes and shrublands showed similar *B. tectorum* densities and higher than wet-meadows (Table 1; Fig. 3A). For instance, the estimated density of *B. tectorum* for shrub-steppes was 8.41 plants/m², 6.82 plants/m² for grassy-steppes, 6.55 plants/m² for shrublands and 3.25 plants/m² for wetlands. Results also show that intermediate and highly disturbed environments present higher densities of *B. tectorum* than those areas with low disturbance (Table 1; Fig. 3B).

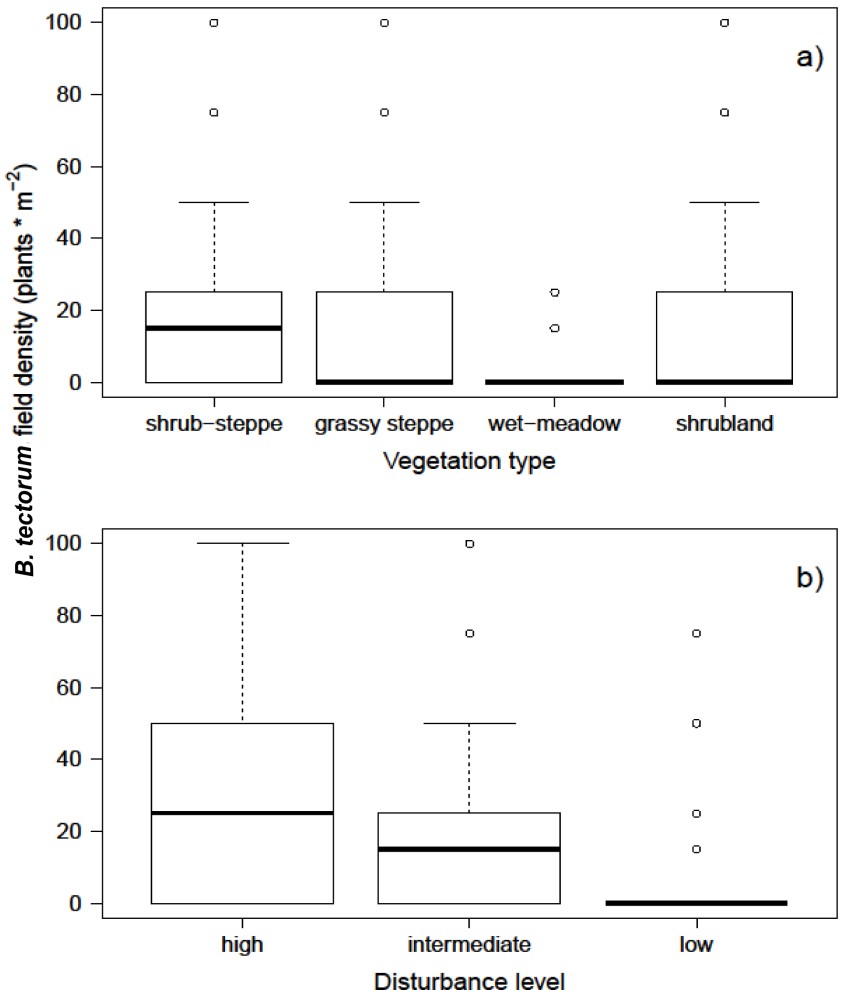

**Figure 3** Observed values of *B.tectorum* density in natural environments. (A) Vegetation type and (B) disturbance level measured in the natural environment.

## DISCUSSION

Our results highlight the importance of road verges on *B. tectorum* invasion into the surrounding native communities. This importance is higher in cases were natural habitats are better conserved and decreases when those habitats are more deteriorated. Previous studies also show that invasive species progressively spread from initial introduction areas as roadsides (*Tyser & Worley, 1992*; *Gelbard & Belnap, 2003*). Roads may be acting in Patagonia as conduits for the spread of *B. tectorum* seeds as in the USA (*Gelbard & Belnap, 2003*), where road maintenance and vehicles contribute to human aided dispersal of invasive species seeds at short and intermediate distances (*Rauschert, Mortensen & Bloser, 2017*; *Rew et al., 2018*). However, their importance depends on disturbance level. We found that density of *B. tectorum* in road verges is more important in determining *B. tectorum* invasion in the surrounding habitats at low levels of disturbance. However, when disturbance is high, the *B. tectorum* density may be explained by self-maintaining
*B. tectorum* populations where propagule source is not the road verge but the populations themselves. However, our data are robust, showing that effects of roads and disturbance are not independent nor additive. Instead, the combined effects among these global change drivers affect *B. tectorum* invasion in north-western Patagonia.

We found a similar *B. tectorum* density in steppes, shrub-steppes and shrublands in the native environments adjacent to the roads which highlights its adaptability and capacity to invade multiple ecosystems in Patagonia. *B. tectorum* germination, seedling and root development is very fast in USA (*Hulbert, 1955*) what probably favours its presence in a constantly human-modified area as roads verges and its success in invading close surrounding ecosystems. Our result contrasts a previous study carried out in the same region that showed that steppes were the most invaded habitat types (*Speziale, Lambertucci & Ezcurra, 2014*). This could represent a change in the invasion pattern with time. However, this previous study was conducted in areas as little disturbed as possible (further away from roads), avoiding wet-meadows, and did not separate steppes from shrub-steppes. The special case of wet-meadows probably represents a niche outside *B. tectorum* ecological needs given that these edaphic community soils are normally wetter than this species requirements and with high vegetation cover (*Utrilla, Brizuela & Cibils, 2005*), being probably environments where *B. tectorum* cannot outcompete native vegetation.

The decline we found in the density of *B. tectorum* with distance from roads for all habitat types evidences the importance of roads during the first stages of the invasion. At the same time, it points at a greater competitive ability of native species, differences in disturbance at lower scale than we measured or in native species composition, and/or at a lack of *B. tectorum* propagules which have probably not reached the interior yet. Competitive ability and seed dispersal at medium distance often account for invasion success (*Brandt & Rickard, 1994*). Well conserved local communities might be dominated by competitive species that are able to outcompete *B. tectorum* in areas with low levels of disturbance (*Davis, Grime & Thompson, 2000*; *Fridley et al., 2007*). However, we found *B. tectorum* close to the road in all the habitats, with diverse disturbance level, and a decline in density as we move away from them. Despite we selected homogenous habitats it is possible that there were differences within plots in disturbance or community's composition that we did not measure. Additionally, dispersal ability could explain the decreased density further away from roads. *B. tectorum* can disperse only one or two metres without the aid of a vector. Its short dispersal ability could be compensated if the presence of cattle would have aid the dispersal of propagules given their adaptations to epizoochory (*Hulbert, 1955*; *Mack, 1981*). Accurate measures of dispersal distance through wind and cattle vectors are not available. This remains to be tested as no study has analysed *B. tectorum* dispersal and/or competitive ability with native species, nor herbivory and seed predation in Patagonia. Alternative, *B. tectorum* may be just ending a latency period given its relatively short time since its first introduction in Patagonia (*Biganzoli, Larsen & Rolhauser, 2013*; *Speziale, Lambertucci & Ezcurra, 2014*) what could explain its progressive increase in density within the native surroundings.

## CONCLUSIONS

Synergies among global change drivers can impact native environments favouring the invasion of *B. tectorum*. Habitat type is not the only important factor in resisting a *B. tectorum* invasion. Within habitats susceptible to invasion (all but wet-meadows) road verges influenced the density of *B. tectorum* in the surrounding habitat but their importance changed with the habitat disturbance level. This synergy among global drivers needs to be taken into account to develop accurate management tools (*Didham et al., 2005*). According to this synergy, minimizing road construction and improving existing roads, as well as designing road verge vegetation programs to quickly detect and prevent invasions, are important recommendations for the sustainable management of ecosystems (*Tyser & Worley, 1992*; *Gelbard & Belnap, 2003*). Also, from our results stem new recommendations. Given the low invasion of habitats with low sign of degradation, the common sense recommendation would be to keep this good habitat conservation. However, it is important to design and implement both strategies at a time: *B. tectorum* controls in road verges and avoiding high disturbances within the ecosystems. Additionally, when habitat disturbance level is already high, restoration of the ecosystem would be needed in addition to road verge management. This is important both for reducing potential *B. tectorum* invasion impacts but also from a productive point of view which seeks good pastures for the domestic animals.

### Funding

This work was supported by the National Research Council of Argentina (PIP No. 0758-2014), and the ANPCYT of Argentina (PICT No. 2072-2015). The funders had no role in study design, data collection and analysis, decision to publish, or preparation of the manuscript.

### Grant Disclosures

The following grant information was disclosed by the authors:
National Research Council of Argentina: PIP No. 0758-2014.
ANPCYT of Argentina: PICT No. 2072-2015.

### Competing Interests

The authors declare there are no competing interests.

### Author Contributions

- Karina L. Speziale conceived and designed the experiments, performed the experiments, contributed reagents/materials/analysis tools, prepared figures and/or tables, authored or reviewed drafts of the paper, approved the final draft.
- Agustina di Virgilio conceived and designed the experiments, performed the experiments, analyzed the data, contributed reagents/materials/analysis tools, prepared figures and/or tables, authored or reviewed drafts of the paper, approved the final draft.

- Maria N. Lescano, Gabriela Pirk and Jorgelina Franzese conceived and designed the experiments, performed the experiments, contributed reagents/materials/analysis tools, authored or reviewed drafts of the paper, approved the final draft.

## Field Study Permissions

The following information was supplied relating to field study approvals (i.e., approving body and any reference numbers):

Field work was approved by the National Park Administration (project number: 1526).

## Data Availability

The raw data are provided in Supplemental File.

## Supplemental Information

Supplemental information for this article can be found online at http://dx.doi.org/10.7717/peerj.5529#supplemental-information.

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
