# Peer review of "Synergy between roads and disturbance favour Bromus tectorum L. invasion"

_PeerJ, doi:10.7717/peerj.5529_

## Round 0.1 · original submission · Major Revisions

Reviewers are presenting very detailed reports on your manuscript that hopefully could help you in highlighting the quality of your work.

In particular, according to Rev#2 and Rev#3 some conceptual issues should be introduced/discussed more in deepness, including the role provided by roads in the frequency and abundance of B. tectorum and the possible resistance of adjacent habitats to Bromus colonization.

Reviewer 1 ·

Basic reporting

In this manuscript the authors tested the effects of road verges as contributor for the spread of an introduced grass species into surrounding environments. In particular, they tested whether the invasion success in the surrounding native communities depended on the level of propogule pressure in road verges and the level of disturbance in adjacent habitats.

The manuscript is generally well written, however, English should be improved in some parts (e.g. L 71-76, L 88, 203-205, 221-222, 241,...)

Literature references and sufficient field background/context are provided in most places, with some exceptions noted below:

Dainese M, Aikio S, Hulme PE, Bertolli A, Prosser F, Marini L (2017) Human disturbance and upward expansion of plants in a warming climate. Nature Climate Change 7: 577-580
Lembrechts,J.J.etal.Mountain roads shift native and non-native plant species ranges. Ecography 40,353–364(2017).
Lembrechts,J.J.etal.Disturbance is the key to plant invasions in cold environments. Proc.NatlAcad.Sci.USA 113,14061–14066(2016).

The strucuture of the manuscript is OK.

Experimental design

The research question was well defined and is relevant to the expanding field of understanding biological invasions. The study is designed well, methods were detailed and appropriate, and allowed robust analysis of the results.

Minor points:
L. 167-170: Although the analysis included interaction terms, this was not clearly stated in the description of the model.
L 172-173: Why did you use choose these reference groups?

Validity of the findings

The results were presented clearly, expected those related to Fig. 2 (see L 184-188). Observing the fitted lines in Fig. 2, I have some difficulties to outline this result as described by the authors. It's much clear observing the observed values. In this form, Fig. 2 is misleading. Anyway, conclusions were supported by the results. I would only suggest to develop the interpretation of this result (Fig. 2) a little bit (see L 212-218). Another explanation could be that in fields with high levels of disturbance local communities are already dominated by very competitive species that are able to outcompete with cheatgrass. This could explain the independent effect of road verges on cheatgrass density in moderate or highly disturbed habitats.

Additional comments

In my point of view the manuscript need some minor adjustments only, and, overall, the research is well designed and presented good results. I recommend the publication in PeerJ.

Reviewer 2 ·

Basic reporting

This manuscript reports on patterns of Bromus tectorum presence and abundance at different distances to road and in response to habitat type. Overall the results seem robust, and confirm what we understand about B tectorum distribution in semi-arid steppes around the world. That is B tectorum as a ruderal winter annual responds positively to many types of disturbance that often occurs along roads and in areas that are overgrazed by livestock. The most interesting and unique aspect is the inclusion of roadside population size in predicting sites abundance away from the roadside; but this needs some clarification.

Experimental design

Overall the experiment is well designed to detect the patterns of abundance or presence-absence on the Patagonian landscape. The inference to separate the effect of propagule pressure (dispersal), or disturbance seems somewhat limited as currently presented.
There is a positive correlation between the abundance at the roadside and abundance as distance from road decreases, the authors present a model (Table 1), but its unlcear if this model is for all data or just the subset away from the road. I wonder how roadside abundance was included in the model if the roadside data was included as a predictor variable, wouldn’t the correlation of the response and predictor affect the robustness of the model?

Validity of the findings

The findings support the idea of greater occurrence of at the roadside, and that there is a correlation between population density at the roadside and at distances further away. Some of the model is not well explained as mentioned above and I couldn’t understand how the model was fit. Why were the models fit as general additive models in the end and what are coefficients being presented in Table 1, it is not clearly stated.

Additional comments

I found the introduction and discussion could be improved by more literature on B tectorum invasion (See a list below). I also feel the title is not informative and doesn’t not convey what paper is about. I feel all mentions of ‘drivers of global change’ could be deleted and instead use a more explicit term like disturbance or propagule pressure interact to influence B tectorum invasion. I think it would increase the impact and visibility of the manuscript. By deleting these mention the major revisions to the text are needed.
Overall the manuscript reads well, but there is a large number of minor changes to make to language for clarity. I present a list below but it not exhaustive.
Title: Does not relate to actual manuscript, revise to be more direct.
Abstract
I would use only Bromus tectorum in the abstract, and mention in the intro it is also commonly referred to as downey brome
L22 totaling approximately 400 km, better than travelling ca
L33-35; L34, this reads funny, clarify by making two clauses that some where the same and that wetlands had the lowest amount.
L37 synergy is not very informative – interaction of population/propagule pressure habitat and disturbance?
L40 reorder: with healthy environ being .
L45 not a very powerful or informative sentence as the last sentence of the abstract.
Introduction
L49 native species is redundant delete.
L79 These
L86-88 delete these sentence ending with Hulbert citation
I found the last paragraph of the intro needs improvement. Is the last sentence even testable there is a correlation with high disturbance and high propagule pressure how would you separator these results.
Methods
L140 – dirt or gravel road – I suggest using unpaved as a catch all term
L158 states 5 four listed.
L161 feces misspelled
L168 what are co variables? Independent or predictor variables?
L173 label the intercept with reference level in the table. So is the table then the contrast and not the parameter estimates? That should certainly be clarified. I don’t get why a gam was fit in the end, and what is reported in the figure the raw data or the mixed effects fixed parmeters? There was never any mention of spatial autocorrelation in the results or if it was necessary to include.
L184 latter habitats is confusing, rewrite to better reflect you are talking about the only roadsides.
L190 space
L199 fix standard deviation parentheses
L204 roadside, delete that, delete the, better word than favour, promote maybe. How did you separate the effect of disturbance and distance from road? Needs to be addresses if is currently unclear.
L208 Can you really draw that conclusion? They are good habitat but are they influencing spread if disturbance away from the road is a more important factor. Wouldn’t land use in general be a better explanatory variable?
L216-217 addivite and combined effects used the two sentences are confusing, please clarify
L219 The decline in abundance as distance to road increases is better than decay.
L221-222 I feel this is a stretch didn’t it differ by disturbance too? So it is the interaction. A graph that had B tectorum on the y axis and distance to road on the x with lines for each habitat and or disturbance intensity would better convey the results of the study. In fact dropping the wetland habitat would allow a better test of the effect of disturbance, population size, and distance to road.
I felt the conclusion need to be rewritten, were too long, I already stated global change drivers is uninformative, and also the last sentence isn’t directly connected to the paper or results.
Graphs – spell out Bromus tectorum in all axis labels, make the habitat boxes narrow (is that the raw data or fitted data?) The road density field density model is poorly fit where is the last red dot, and black dot does come close to fitted curve. Are these over extrapolated curves?
Tables The caption is uninformative and did state what kind of model it was or what the response variable was.

Reviewer 3 ·

Basic reporting

I have read this manuscript several times and believe the data tells an interesting story, but I found the current framework for the story confusing, and the graphics and analysis are not clear. With some major revisions this will be a useful contribution to the literature.
There are three questions addressed clearly by the data from what I understand. 1) what role do roads, and surface type, play in the frequency and abundance of B. tectorum. (It is unclear if this question can include disturbance intensity at the roadside). 2) How resistant are adjacent habitats to B. tectorum invasion, and 3) is this correlated with disturbance intensity. I believe the final paragraph of the introduction would benefit from a clearer articulation of the study questions/objectives.
The introduction references global change several times, stating land use change (e.g. road construction) and invasions as key ones. If this truly is of interest to the authors they should evaluate age of road in their model, and regardless of that fact type of road should be included in the model. Thus, overall I find the introduction interesting but don’t feel it really clarifies the purpose of the study. The reference to global drivers confuses rather than clarifies plus it’s questionable if the results provided overlap with land use change (road but not habitats as described). There are also quite a few newer publications pertaining to roads as vectors that would be beneficial to include. I will provide a list below.

Overall this manuscript has potential but the story need to be framed more clearly, and the data analysis and presentation more thorough.
Potentially useful references
Balch, J. K., B. A. Bradley, C. M. D'Antonio, and J. Gomez-Dans. 2013. Introduced annual grass increases regional fire activity across the arid western USA (1980-2009). Glob Chang Biol 19:173-183.
Bradley, B. 2012. Distribution models of invasive plants over-estimate potential impact. Biological Invasions:1-13.
Bradley, B. A. 2009. Regional analysis of the impacts of climate change on cheatgrass invasion shows potential risk and opportunity. Global Change Biology 15:196-208.
Bradley, B. A., C. A. Curtis, and J. C. Chambers. 2016. Bromus Response to Climate and Projected Changes with Climate Change.257-274.
Mortensen, D. A., E. S. J. Rauschert, A. N. Nord, and B. P. Jones. 2009. Forest roads facilitate the spread of invasive plants. Invasive Plant Science and Management 2:191-199.
Rauschert, E. S. J., D. A. Mortensen, and S. M. Bloser. 2017. Human-mediated dispersal via rural road maintenance can move invasive propagules. Biological Invasions:10.1007/s10530-10017-11416-10532.
Rew, L. J., T. J. Brummer, F. W. Pollnac, C. D. Larson, K. T. Taylor, M. L. Taper, J. D. Fleming, and H. E. Balbach. 2018. Hitching a ride: Seed accrual rates on different types of vehicles. Journal of Environmental Management 206:547-555.
Vakhlamova, T., H.-P. Rusterholz, Y. Kanibolotskaya, and B. Baur. 2016. Effects of road type and urbanization on the diversity and abundance of alien species in roadside verges in Western Siberia. Plant Ecology 217:241-252.
von der Lippe, M., and I. Kowarik. 2007. Long-distance dispersal of plants by vehicles as a driver of plant invasions. Conservation Biology 21:986-996.
von der Lippe, M., and I. Kowarik. 2012. Interactions between propagule pressure and seed traits shape human-mediated seed dispersal along roads. Perspectives in Plant Ecology, Evolution and Systematics 14:123-130.

Experimental design

The methods are unclear. I cannot make 88 plots divisible by sampling every 5 m and the distance given. A table with number of samples per road – 14 I assume though I think that means a greater overall distance. Is there much elevation difference – I assume not given the location but stating this would be useful. How many samples were in each habitat type? These data should be provided.
How far was the interior transect from the road – a minimum, mean and maximum may be useful here. Was disturbance recorded along the roadside? This seems like an important variable.

Validity of the findings

From the results it appears comparisons were made of infestation levels by road type. I don’t believe this is mentioned in the analysis section.
Why choose the intermediate as the mid point, and the shrubland. Were these the most populous?
Fig 2. The lines on the graph do not correspond with the error bars, nor really the model. Is this correct? Please check – it is not convincing as presented. Furthermore, having road type in the model seems vital if it was significant from other analysis that isn’t show – or is it highly correlated with the abundance variable? Wouldn’t it be more informative for management to use road type not abundance if they are correlated? It would agree with recent results from Europe.
For Fig 3 what happens if the 100 m is removed from analysis, is there still a decline? In how many cases was B. tectorum observed more than 100 m. Several manuscripts I have read recently have used the mid-point of the density group, or better generated random number generation within each group. Was this considered, does it improve the model? The figure would be better with the raw data displayed as scatter points with the model prediction, rather than the SE bars.
The results and discussion suggest the study was more interested in habitat resistance. If this is true then some references are missed.
The first sentence of discussion is confusing, and I don’t know what to suggest. Later, line 214, as well as other places I was often confused if disturbance refers only to the interior habitat. Was it not measured at the roadside? That seems so important.
Line 22 – be careful talking about interior - 50 m is hardly very far.
Paragraph 229-240. This paragraph almost contradicts itself with regard to dispersal, and there are studies that could be cited that are more relevant (see below).
Line 241. Very confusing. The rest of the paragraph discussed data that I do not find in the graphs and tables, nor methods, though it is mentioned in the results. See Rauschert for another study on seed movement.

Additional comments

Overall this manuscript has potential but the story need to be framed more clearly, and the data analysis and presentation more thorough.

---

## Round 0.2 · Minor Revisions

Some minor changes are still requested by Reviewer 3. Please, consider in particular his/her suggestions about the preparation of the figures, in order to make them more effective in supporting your results.

Reviewer 3 ·

Basic reporting

The article is fairly well written, there are some inconsistencies that need addressing.

Experimental design

Clearer now.

Validity of the findings

Clear, figures could be improved.

Additional comments

Synergy between roads and disturbance favour Bromus tectorum invasion
Speziale et al.
Revision 1.
I thank the authors for their improvements to the manuscript, there are now only some minor errors to correct.
The number of sample points is still unclear. It says 88 in the abstract on line 22, 73 on line 146 and 74 in the response to reviewers.

Abstract:
Line 18, suggest the degree of invasion, rather than what you have
Line 25 suggest In not From, and recorded not registered
Line 30. Unclear. Perhaps put the second sentence first.
Line 40 “healthy” - intact may be more appropriate word
Line 69 – yes, roadsides can be composed of many short-lived fast growing species, but in my experience this is rare on at least three continents. Perennial grasses, sown or not, are more the norm so I would suggest mediating this statement even if you don’t have a citation.
Line 90 downy doesn’t have an e. I am not sure what this journal prefers but in this paragraph you have B. tectorum after the first mention but in other places you use cheatgrass. Be consistent and determine that the journal requires.
Line 92 consider adding “and degraded rangelands” to your sentence, this was found by several of your cited references and helps your arguement.
Line 106 consider widespread not notorious
Line 107 consider with not being, and a comma after persistence
Line 11 consider is not was
Line 129 Spelling of shrublands
Line 144 – units and formatting are all a muddle e.g. kilometres, km, m with and without a space. Check formatting and consistency here and elsewhere.
Line 165 disturbance
Line 169 no Title case in other places so I would drop.
Line 230 While it can germinate in the spring it is really a fall germinating species.
Figures
Please consider that red does not come out differently on a black and white printer so you may wish to alter the colour or pattern.
Fig 2. I still do not find this figure useful because of the poor fit – especially for the low disturbance which looks linear. The authors seem resolved not to alter the presentation but I would suggest it would improve the citation capacity of the paper.
For all figures (BT) is not needed as it is not used as an abbreviation anywhere. Again, check the journal requirements but having common in title and scientific name on figure seems messy. There is also a mix of Title Case and Sentence case which needs to be rectified.
Units for density need to be on the figures
Figure 3 should have 0 for y axis

---

## Round 0.3 · accepted · Accept

The requested changes have been fulfilled and your manuscript is now suitable for publication. Thank you and best wishes for your work.

#